# Observations of marine cold-air outbreaks: A comprehensive data set of airborne and dropsonde measurements from the Springtime Atmospheric Boundary Layer Experiment (STABLE)

Janosch Michaelis[1], Amelie U. Schmitt[2], Christof Lüpkes[1], Jörg Hartmann[1], Gerit Birnbaum[1], and Timo Vihma[3, 4]

[1]Alfred Wegener Institute Helmholtz Centre for Polar and Marine Research, Bremerhaven, Germany
[2]Meteorological Institute, Center for Earth System Research and Sustainability (CEN), Universität Hamburg, Hamburg, Germany
[3]Finnish Meteorological Institute, Helsinki, Finland
[4]The University Centre in Svalbard, Longyearbyen, Norway

**Correspondence:** Janosch Michaelis (janosch.michaelis@awi.de)

**Abstract.** In March 2013, the Springtime Atmospheric Boundary Layer Experiment (STABLE) was carried out in the region of Fram Strait and over Svalbard to investigate atmospheric convection and boundary layer modifications due to interactions between sea ice, atmosphere, and open water. A major goal was the observation of marine cold-air outbreaks (MCAOs), which are typically characterised by a transport of very cold air masses from the ice covered ocean over a relatively warm water surface, and which often affect local and regional weather conditions. During STABLE, such MCAOs were observed on four days within a period of a strongly northward shifted sea ice edge north of Svalbard and thus with an unusually large Whaler's Bay Polynya. The observations mainly consisted of in situ measurements from airborne instruments and of measurements by dropsondes. Here, we present the corresponding data set from, in total, 15 aircraft vertical profiles and 22 dropsonde releases. Besides an overview on flight patterns and instrumentation, we provide a detailed presentation of the individual quality-processing mechanisms, which ensure that the data can be used, for example, for model validation. Moreover, we discuss the effects by the individual quality-processing mechanisms and we briefly present the main characteristics of the MCAOs based on the quality-controlled data. All 37 data series are published in the World Data Center PANGAEA (Lüpkes et al., 2021a, https://doi.pangaea.de/10.1594/PANGAEA.936635).

## 1 Introduction

The Springtime Atmospheric Boundary Layer Experiment (STABLE) was an aircraft campaign led by the German Alfred-Wegener-Institut. It was conducted in the Fram Strait region west and north of Svalbard in March 2013. One of the main objectives of the campaign were measurements during marine cold-air outbreaks (MCAOs). MCAOs are characterised by advection of cold air masses typically originating from the sea ice covered ocean over a relatively warm water surface, which can result in moderate or strong convection depending on the season. In the high-latitudes, MCAOs represent one of the strongest events of atmosphere–ocean interaction (Brümmer, 1996). Thus, they often affect local and regional weather conditions, for

example, by promoting polar low formation (Rasmussen, 2003). Particularly in the Fram Strait region, strong Northern Hemisphere MCAO events occur in high frequency (Brümmer and Pohlmann, 2000; Fletcher et al., 2016). Strong MCAOs were also observed during STABLE. On four days, detailed investigations were performed using in situ and remote sensing instrumentation of the research aircraft Polar 5 as well as dropsondes. In this paper, we provide a detailed presentation of the atmospheric measurements in the lower troposphere related to the MCAOs observed during STABLE.

The data set consists of measurements from 15 vertical aircraft profiles (Table 1) and 20 dropsondes (Table 2). Data from two additional dropsondes, which were released to investigate spatial and temporal differences of the observations, are also included (see also Table 2). All data were obtained over the marginal sea ice zone (MIZ) as well as over the open ocean region nearly along the direction of the lower atmospheric flow in the considered MCAOs (see Fig. 1). Each data series consists of quality-controlled measurements of temperature, humidity, wind, and pressure. As the main quality-processing of the aircraft data, we corrected related air temperature measurements for the adiabatic effect of the dynamic pressure originating from the motion of the aircraft. Air pressure measurements were corrected for the influence of the flow field around the aircraft. For the dropsonde data, multiple corrections were applied for which we used the Atmospheric Sounding Processing Environment software (ASPEN, see Martin and Suhr, 2021). The data set is accessible on the World Data Center PANGAEA (Lüpkes et al., 2021a, https://doi.pangaea.de/10.1594/PANGAEA.936635).

Parts of the data set have already been used by Tetzlaff et al. (2014) and Tetzlaff (2016) to study the MCAO development during STABLE in spring 2013. In that period, the MCAO development was - at least in its northern part - extreme due to a northward shift of the ice edge and an unusually large width of the Whaler's Bay Polynya north of Svalbard (Tetzlaff et al., 2014, their Fig. 1). Overall, climate projections suggest, however, a future weakening of MCAOs and, correspondingly, a weaker or reduced polar low development by increased sea ice loss in their source region (e.g., Kolstad and Bracegirdle, 2008; Zahn and von Storch, 2010; Landgren et al., 2019). The Fram Strait region is, in particular, marked by a stronger than average retreat in Arctic sea ice extent (Cavalieri and Parkinson, 2012). As shown by Tetzlaff et al. (2014) and Tetzlaff (2016), it is also a prominent example of an already ongoing poleward movement of an MCAO source region.

In general, MCAOs are accompanied by a large variety of small-scale processes (e.g., roll- and cellular convection, cloud radiative processes, phase changes of water, local and non-local turbulence). In turn, these processes depend on many different preconditions (e.g., sea ice structure and concentration, moisture content above the temperature inversion), which complicates their exact representation in numerical weather prediction and climate models (e.g., Pithan et al., 2018). Observational and modelling studies on the respective processes can be found in, for example, Brümmer et al. (1992); Lüpkes and Schlünzen (1996); Brümmer (1997); Hartmann et al. (1997); Brümmer (1999); Gryanik and Hartmann (2002); Liu et al. (2006); Gryschka et al. (2008); Chechin et al. (2013); Gryschka et al. (2014); Chechin and Lüpkes (2017); Geerts et al. (2021) and more general reviews, e.g., in Etling and Brown (1993); Brümmer and Pohlmann (2000); Lüpkes et al. (2012); Vihma et al. (2014).

With the data set that we describe in this paper, we aim to provide reliable and highly resolved atmospheric measurements. They can be used as a valuable reference for further observational studies as well as for validation of model simulation results. The data refer to observations that cover not only the open ocean region with well-developed convective boundary layers but also the MCAOs' source regions over almost closed sea ice. Moreover, the exceptionally low sea ice concentration north of

Svalbard clearly influenced the measurements so that extreme atmospheric boundary layer (ABL) heights had been observed already far in the North. Thus, the data may be used to better understand the detailed small-scale processes related to MCAOs, and they may also help to understand the future trends of polar air mass transformations.

Section 2 deals with a brief overview of the campaign STABLE, including the large-scale weather situations causing the MCAOs. In Sect. 3, we describe the aircraft's instrumentation and the quality-processing applied to the corresponding measurements. In Sect. 4, we do the same for the dropsondes. In Sect. 5, we briefly describe the horizontal distances covered during the aircraft's vertical flight sections and during the measurements of each dropsonde. In Sect. 6, we describe some characteristics of the MCAOs based on the quality-controlled data. Finally, a data availability statement is made in Sect. 7 and conclusions are drawn in Sect. 8.

**Table 1.** Overview of all aircraft ascents and descents performed mostly upwind of the sea ice margin on 4, 6, and 7 March 2013.[a]

| Date | Flight leg | Start time (UTC) | Mean fetch (km) | Link to data series in PANGAEA |
|------|-----------|------------------|-----------------|--------------------------------|
| | T1 | 12:03:32 | -179 | https://doi.pangaea.de/10.1594/PANGAEA.936639 |
| | T2 | 12:05:40 | -178 | https://doi.pangaea.de/10.1594/PANGAEA.936645 |
| | T3 | 12:53:55 | -290 | https://doi.pangaea.de/10.1594/PANGAEA.936646 |
| 4 March | T4 | 12:56:20 | -289 | https://doi.pangaea.de/10.1594/PANGAEA.936647 |
| | T5 | 13:42:55 | -404 | https://doi.pangaea.de/10.1594/PANGAEA.936648 |
| | T6 | 13:46:45 | -403 | https://doi.pangaea.de/10.1594/PANGAEA.936650 |
| | T7 | 15:04:19 | -64 | https://doi.pangaea.de/10.1594/PANGAEA.936651 |
| | T1 | 12:19:08 | -106 | https://doi.pangaea.de/10.1594/PANGAEA.936655 |
| | T2 | 12:24:15 | -105 | https://doi.pangaea.de/10.1594/PANGAEA.936656 |
| | T3 | 13:09:40 | -217 | https://doi.pangaea.de/10.1594/PANGAEA.936660 |
| 6 March | T4 | 13:13:52 | -215 | https://doi.pangaea.de/10.1594/PANGAEA.936661 |
| | T5 | 13:48:37 | -296 | https://doi.pangaea.de/10.1594/PANGAEA.936663 |
| | T6 | 13:52:48 | -297 | https://doi.pangaea.de/10.1594/PANGAEA.936664 |
| | T7 | 15:05:42 | 33 | https://doi.pangaea.de/10.1594/PANGAEA.936665 |
| 7 March | T1 | 14:50:40 | -13 | https://doi.pangaea.de/10.1594/PANGAEA.936666 |

[a] The mean fetch along the flight tracks denotes the mean distance to the sea ice edge during the flight legs with negative/positive values denoting flights over sea ice/open water. These distances were determined along the 5° E (4 March), 2.5° E (6 March), and 2° E (7 March) meridians, respectively.

## 2  The campaign STABLE

We present a data set that was collected during the campaign STABLE in the Fram Strait region north and west of Svalbard on 4, 6, 7, and 26 March 2013. The large-scale weather patterns on those days promoted the formation of strong MCAOs. The observations on 4, 6, and 7 March all concentrated on the same MCAO episode. In the course of 3 March, northerly winds and thus a strong off-ice flow had developed in the lower troposphere over Fram Strait between a low pressure system east

**Table 2.** Overview of all dropsondes released on 4, 6, 7, and 26 March 2013.[a]

| Date | Dropsonde No. | Release time (UTC) | Mean fetch (km) | Link to data series in PANGAEA |
|---|---|---|---|---|
| | D1 | 15:19:38 | -5 | https://doi.pangaea.de/10.1594/PANGAEA.936612 |
| | D2 | 15:31:19 | 48 | https://doi.pangaea.de/10.1594/PANGAEA.936613 |
| 4 March | D3 | 15:42:07 | 104 | https://doi.pangaea.de/10.1594/PANGAEA.936614 |
| | D4 | 15:53:16 | 161 | https://doi.pangaea.de/10.1594/PANGAEA.936615 |
| | D5 | 16:04:54 | 214 | https://doi.pangaea.de/10.1594/PANGAEA.936616 |
| | D1 | 15:16:25 | 57 | https://doi.pangaea.de/10.1594/PANGAEA.936617 |
| | D2 | 15:28:02 | 109 | https://doi.pangaea.de/10.1594/PANGAEA.936618 |
| 6 March | D3 | 15:38:02 | 165 | https://doi.pangaea.de/10.1594/PANGAEA.936619 |
| | D4 | 15:49:04 | 220 | https://doi.pangaea.de/10.1594/PANGAEA.936620 |
| | D5[b] | 16:06:44 | 92 | https://doi.pangaea.de/10.1594/PANGAEA.936621 |
| | D1 | 15:06:41 | 49 | https://doi.pangaea.de/10.1594/PANGAEA.936622 |
| 7 March | D2 | 15:17:51 | 114 | https://doi.pangaea.de/10.1594/PANGAEA.936623 |
| | D3 | 15:46:46 | 203 | https://doi.pangaea.de/10.1594/PANGAEA.936624 |
| | D1[c] | 11:54:02 | -59 | https://doi.pangaea.de/10.1594/PANGAEA.936625 |
| | D2 | 14:13:31 | -59 | https://doi.pangaea.de/10.1594/PANGAEA.936626 |
| | D3 | 14:30:09 | 35 | https://doi.pangaea.de/10.1594/PANGAEA.936627 |
| | D4 | 14:42:20 | 107 | https://doi.pangaea.de/10.1594/PANGAEA.936628 |
| 26 March | D5 | 14:52:16 | 166 | https://doi.pangaea.de/10.1594/PANGAEA.936629 |
| | D6 | 15:01:26 | 223 | https://doi.pangaea.de/10.1594/PANGAEA.936630 |
| | D7 | 15:10:23 | 277 | https://doi.pangaea.de/10.1594/PANGAEA.936631 |
| | D8 | 15:18:03 | 324 | https://doi.pangaea.de/10.1594/PANGAEA.936632 |
| | D9 | 15:27:28 | 381 | https://doi.pangaea.de/10.1594/PANGAEA.936633 |

[a] The mean fetch denotes the mean distance to the sea ice edge along the flight track for each dropsonde. For 4, 6, and 7 March, these distances were determined along the $5^{°}$ E (4 March), $2.5^{°}$ E (6 March), and $2^{°}$ E (7 March) meridians, respectively. For 26 March, the values refer to the sea ice edge at $80.9^{°}$ N and $17^{°}$ E, which approximately marks the northeasternmost extension of the Whaler's Bay Polynya on that day (see Fig. 1d). [b] This dropsonde was released at the same latitude as dropsonde D4 from the same day but more to the west and with shorter fetch. [c] This dropsonde was released almost at the same position as dropsonde D2 from the same day but approximately 2:20 h earlier (see also Fig. 1).

of Svalbard and a strong high pressure system over Greenland. These large-scale weather conditions more or less persisted for another four days although on 7 March in the southern part the lower tropospheric flow slightly turned to northwest (Fig. 1a–1c). On 8 March, the MCAO weakened due to a high pressure ridge that moved to the region west of Svalbard. The satellite images in Fig. 1 show that especially on 26 March the MCAO affected the Whaler's Bay Polynya. On that day, the flow had almost a northeast–southwest orientation (Fig. 1d) and was thus directed along the corresponding axis of the polynya. The corresponding large-scale weather pattern was characterised by a low pressure system located at the southern tip of Svalbard, which had developed in the previous night.

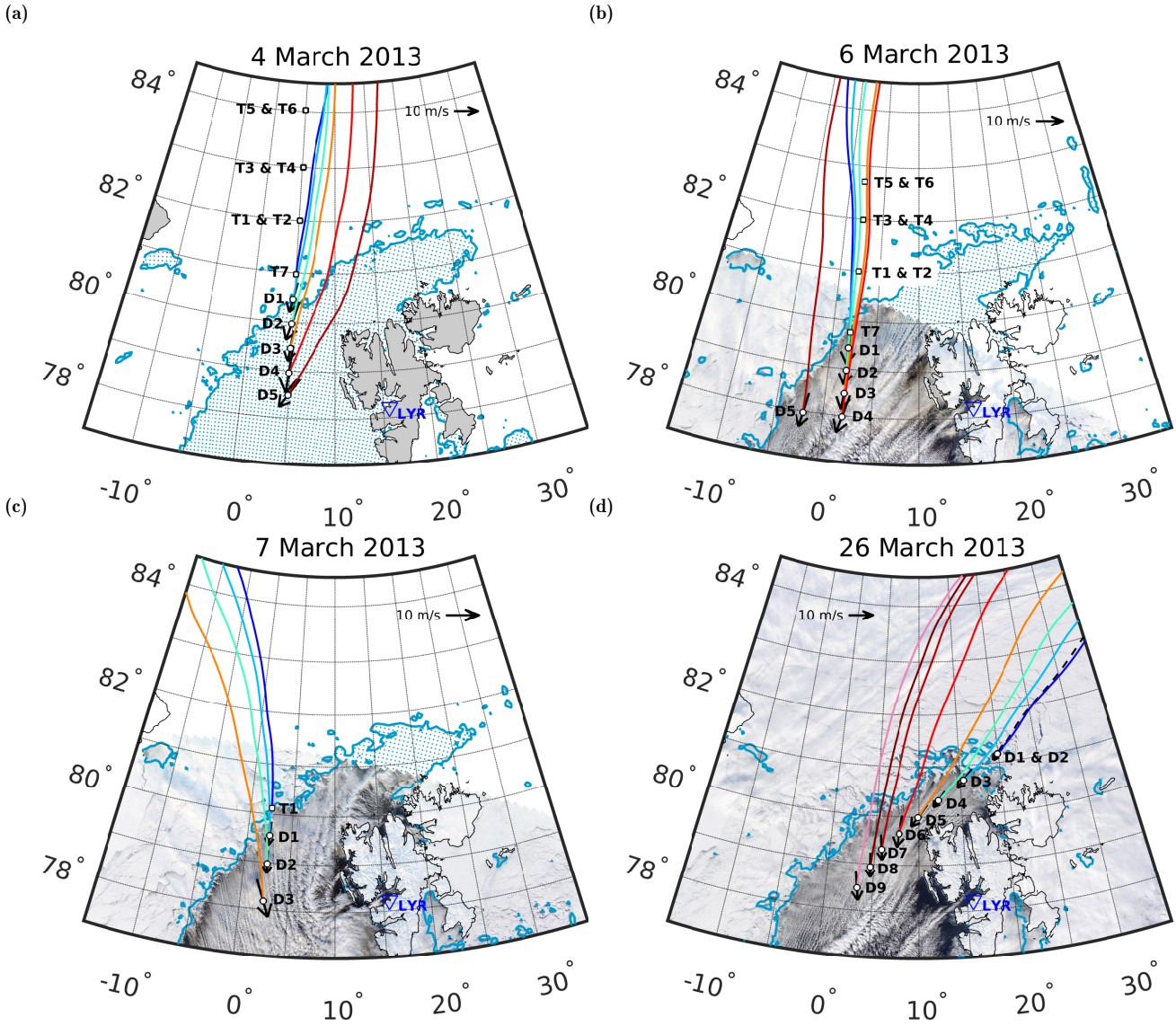

**Figure 1.** Ice edge based on a 70 % threshold value of the ice concentration (derived with the ARTIST Sea Ice algorithm for the Advanced Microwave Scanning Radiometer 2, see Spreen et al., 2008) during MCAOs on (a) 4 March, (b) 6 March, (c) 7 March, and (d) 26 March 2013. The arrows denote the vertically ABL-averaged wind at the positions of the dropsondes listed in Table 2. In (a)–(c), locations of the aircraft profiles listed in Table 1 are shown. Coloured lines are backward-trajectories obtained with the Hybrid Single-Particle Lagrangian Integrated Trajectory transport and dispersion model (Draxler and Rolph, 2010) at 10 m height from the positions of the dropsondes and of the southernmost aircraft vertical profiles for 4, 6, and 7 March. Backgrounds of (b)–(d) are the corresponding MODIS visible images at 13:10 UTC, 12:15 UTC, and 12:45 UTC, respectively (data from https://lance.modaps.eosdis.nasa.gov/modis/). Blue triangles mark the location of Longyearbyen airport (LYR). Modified based on Tetzlaff et al. (2014) and Tetzlaff (2016).

The measurements during STABLE were performed with the Polar 5 research aircraft, a Basler BT-67[1]. The following description of flight patterns and of the aircraft's instrumentation in Sect. 3.1 is partly based on Tetzlaff et al. (2014) and Tetzlaff (2016). Apart from observing MCAOs, an additional topic of STABLE was the measurement of ABL convection over leads in sea ice (mean and turbulent quantities). These measurements are described in detail by Tetzlaff et al. (2015), Tetzlaff (2016), and Michaelis et al. (2021), and the corresponding data can be found in Lüpkes et al. (2021b).

## 2.1 Flight patterns

All research flights of typically 5–6 hours duration started and ended at the airport of Longyearbyen (see also Fig. A1 in Appendix A). The measurements described in this paper took place between approximately 12:00 and 16:00 UTC on each day. Horizontally, they covered an area between approximately 78–84° N and -5–25° E (Fig. 1). Basically, the flights were organised as follows. Especially over the open ocean region with strong convective rolls and related clouds, flights were carried out at constant altitude at about 3000 m. There, the dropsondes were released. Prior to this, flight sections were flown north of the sea ice edge mainly at low levels in the ABL but some of them also at 3000 m. These sections were interrupted at some points by ascents and descents to measure the vertical structure of the ABL. Cloud cover decreased during all flights towards North, while in turn the sea ice concentration increased to more than 90–95 %. The data set we describe here was gained from the ascents and descents as well as from the dropsondes (see Fig. 1 and Tables 1–2).

For both 4 and 6 March, the data set consists of measurements from seven ascents and descents over the MIZ northwest of Svalbard (T1–T7) and of five dropsondes (D1–D5). On both days, the aircraft profiles T1–T6 were performed between about 40 m height and an altitude of 500–1000 m, which was above the shallow ABL. For these flight legs, the lower limits of the profiles were mostly determined by the low-level cloud conditions over sea ice. The remaining flight legs T7 represent ascents performed up to almost 3000 m. For 7 March, measurements are available only from one ascent over the sea ice edge (T1) and from three dropsondes (D1–D3). For 26 March, only dropsonde measurements are included (D1–D9).

All dropsondes were released mainly over the open ocean region between the sea ice edge and a few 100 km downwind. Only three sondes were used north of the sea ice edge and thus over the sea ice covered region (D1 from 4 March as well as D1 and D2 from 26 March). On 4, 6, and 7 March, the ascents, descents, and dropsonde measurements were performed along a certain meridian. The meridional orientation of the flight patterns roughly corresponded with the main ABL flow direction in the MCAO as shown by the ABL-averaged winds at the dropsondes' positions (Fig. 1a–1c). The corresponding MODIS visible images (for 6 and 7 March) denote that the main orientation of the cloud streets agreed well with the ABL-averaged winds on 7 March. However, a small shift is shown for 6 March, which might be due to the 3 h time difference between the satellite image and the dropsonde measurements (Fig. 1b). The MCAO observed on 26 March originated over the Whaler's Bay Polynya so that the dropsonde measurements were performed along the main MCAO orientation from northeast to southwest (Fig. 1d).

---

[1]for more information, see https://www.awi.de/en/expedition/flugzeuge/polar-5-6/artikel/retrofitted-into-a-polar-research-aircraft.html

## 3 Aircraft measurements

### 3.1 Instrumentation

The data from the aircraft vertical profiles consist of high-frequency meteorological measurements and of GPS- and Internal Navigation System (INS)-based measurements for position and altitude. Altitude is also provided based on the measured atmospheric static pressure. The corresponding offset in the original data, caused by the difference between the reference pressure at departure and the local near-surface pressure, was corrected using radar altimeter measurements during low-level flight legs between the vertical profiles. Differences between pressure and GPS height are explained in more detail in Sect. 3.2. Meteorological quantities were measured by instruments installed in and at the aircraft's nose-boom. Air pressure was measured with a five-hole probe, where the individual pressure components were then used to derive horizontal and vertical wind components. Note that we provide here only mean meteorological quantities since the systematic determination of turbulent fluxes would have required different flight patterns. Vertical wind is only available as a deviation from the average value during horizontal flight sections (see also Ehrlich et al., 2019) so that this quantity is also not provided in the final airborne data set. Temperature was measured with a Pt100 resistance thermometer, and relative humidity with a dew point mirror. Pressure and temperature sensors responded fast enough to obtain a recording frequency of 100 Hz, whereas for humidity it is lower (1 Hz). In the final, quality-controlled data set, all measurements are provided with 100 Hz resulting in a vertical resolution of approximately 5 cm. Measurement accuracies are determined as $\pm 0.1$ hPa for pressure, $\pm 0.01$ K for temperature, $\pm 0.4\%$ for the relative humidity readings, and $\pm 0.2\,\mathrm{m\,s^{-1}}$ for the horizontal wind components (Hartmann et al., 2018). Note that especially the accuracy of humidity measurements depends on environmental conditions. The given accuracy for the wind components is valid rather for horizontal flight legs and might be worse during vertical profiles due to a different pressure field surrounding the aircraft (Hartmann et al., 2018). More details on the aircraft's instrumentation, the individual sensors, calibration procedures, and on the accuracies of the resulting data are provided in Hartmann et al. (2018) and Ehrlich et al. (2019, their Sect. 3.1).

### 3.2 Quality-processing

Post-flight quality-controlling for the airborne measurements included the interpolation of GPS- and INS-data to the time of the sensors installed at the turbulence nose-boom. In addition, although there was not a standard procedure to remove spikes or outliers after the basic processing as described in Hartmann et al. (2018) had been applied, all data series shown here were inspected visually. Sections of aircraft data where invalid values have been identified, for example, due to the influence by icing, are not included in the data stored in the repository.

Air pressure data were corrected for the influence of the flow field around the aircraft. The corrected (static) air pressure $p_s$ was obtained via

$$p_s = p_{s,i} + q_i(1-c) + \Delta p_s, \tag{1}$$

where $p_{s,i}$ is the uncorrected static pressure, $q_i$ is the uncorrected dynamic pressure, $c$ is a calibration factor to obtain the corrected dynamic pressure (with $c = 1.165$), and $\Delta p_s$ is the measurement error of the five hole probe depending on the flow

angle (see Hartmann et al., 2018, their Eq. (6)). The constant $c$ had been obtained by Hartmann et al. (2018) from several pairs of reverse-heading flight sections during which the mean wind had changed only a little.

The wind components were calculated by the difference between the aircraft's velocity and the vector of the true airflow following the method described in detail by Hartmann et al. (2018). While the former component was obtained with a high accuracy from the GPS and INS, the latter was obtained from the quality-controlled pressure measurements. Thus, for the wind components, there was no additional quality-processing necessary apart from the manual check of all time series from which invalid sections had been removed. This held also for the humidity measurements.

Air temperature data were corrected for the adiabatic effect of the dynamic pressure originating from the motion of the aircraft. The following formula was applied:

$$T = (Te_N + 273.15\,\mathrm{K}) \cdot (p_s/(p_s + q_c))^{R/c_p} - 273.15\,\mathrm{K}, \tag{2}$$

where $T$ is the corrected air temperature, $Te_N$ is the temperature measured by the sensor in °C, $q_c$ is the quality-controlled dynamic pressure, $R$ is the specific gas constant of dry air, and $c_p$ is the specific heat capacity of dry air at constant pressure. According to Eq. (2), temperatures are lower after the correction. For our aircraft data set, the average difference between uncorrected and corrected temperatures is 2–3 K. The maximum difference among all flight legs is almost 5 K.

In Fig. 2, we show the differences reported between the aircraft's altitude measured by GPS and as derived from the recorded and quality-controlled static pressure for each vertical profile. Mostly, the two data series differ by less than 10 m. Larger deviations are shown especially for some profiles of 4 March, peaking at a difference of more than 30 m on average for the profile T6. For most profiles, the values measured by GPS were higher than the pressure-based values. This holds also for the height measurements of the dropsondes (see Sect. 4.2.1 and 4.2.3).

## 4 Dropsonde measurements

### 4.1 Instrumentation

The dropsondes of type RD93 manufactured by Vaisala were launched from the aircraft using the Airborne Vertical Atmosphere Profiling System (AVAPS, software version 1.7.1, see Ikonen et al., 2010) in its lite version, which can process only one dropsonde at a time (Vaisala, 2009). Here, we briefly summarise the main properties of both the sondes and the AVAPS based on the more detailed information given by Hock and Franklin (1999), Vaisala (2009), and Ikonen et al. (2010). Each sonde has a diameter of 7 cm, is 41 cm long, weighs about 390 g, and is equipped with temperature, humidity, and pressure sensors. Position, height, as well as wind speed and direction are derived from a GPS module that can track up to 12 satellites simultaneously. Pressure, temperature, and humidity measurements are collected by the AVAPS every 0.5 s and GPS-derived wind every 0.25 s. The AVAPS reported also the number of satellites used for the individual GPS-based measurements and the wind error. Following Hock and Franklin (1999), the latter is determined based on the measurement errors in the sondes' horizontal and vertical velocities and accelerations. The latter error consists of a random component (i.e. noise in the velocity estimates) and of a sampling component due to the sampling interval of the wind measurements (see Hock and Franklin, 1999).

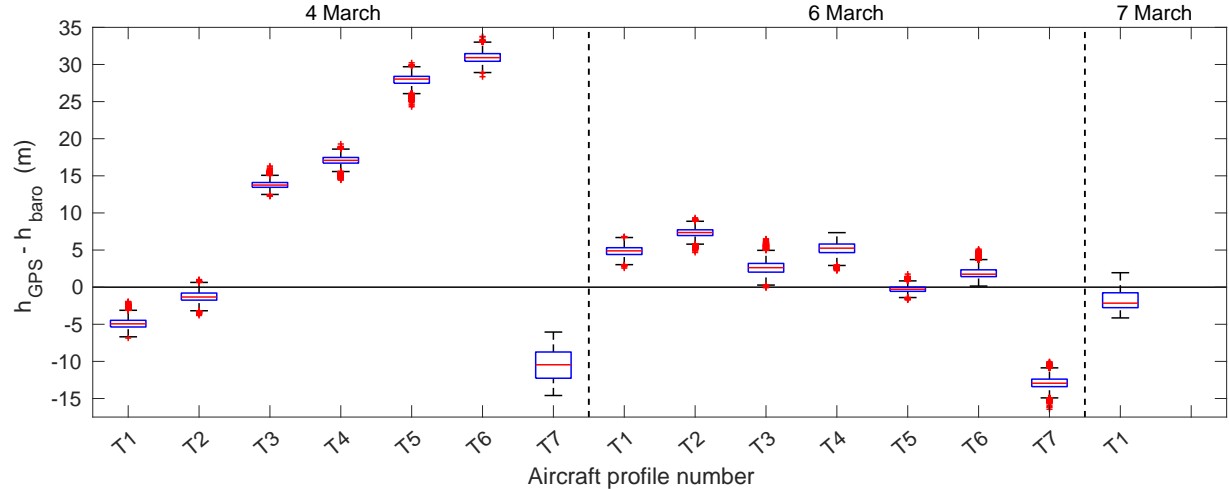

**Figure 2.** Distribution of the differences between the aircraft's height measured by GPS ($h_{GPS}$) to the corrected pressure altitude ($h_{baro}$) for each aircraft profile (see Table 1). Red lines show the median, the boxes are bounded by the upper and lower quartiles, the whiskers include the interval of approximately 99.7 % of the values, and red crosses denote outliers.

Considering the sondes' fall rates of about -10 m s$^{-1}$ during STABLE, a vertical resolution of approximately 2.5 m was obtained for wind and 5 m for the other meteorological quantities. Measurement accuracies are indicated by the manufacturer as 0.2 K for temperature, 0.4 hPa for pressure, 2 % for relative humidity ($\pm 2$ % relative humidity for all readings), and 0.5 m s$^{-1}$ for the horizontal wind speed (Vaisala, 2009). Following the remarks given by Hock and Franklin (1999), these values might be larger in non-laboratory conditions. For an older version of the Vaisala GPS dropsondes, they estimated measurement errors of 0.2 K for temperature, 1.0 hPa for pressure, $< 5\%$ for relative humidity, and 0.5–2 m s$^{-1}$ for the horizontal wind speed (Hock and Franklin, 1999, their Table 3). As mentioned above, the magnitude of the latter error depends on the uncertainties in the position measurements via GPS. The measurements transmitted by each dropsonde were stored by the AVAPS as ascii-formatted text files, for which we then applied quality-processing mechanisms using ASPEN.

## 4.2 Quality-processing

The need to apply quality-processing to the dropsondes' raw data of the campaign STABLE was shown by Tetzlaff (2016). They compared measurements of a dropsonde released over the MIZ with nearby airborne measurements obtained during an ascent and a descent. As stated by Tetzlaff (2016), the airborne measurements act as a useful reference for the dropsonde data due to the dropsonde's lower resolution and simpler instrumentation. The validation by Tetzlaff (2016) was performed over a region with 90–95 % sea ice cover on 20 March 2013. On this day, a stable boundary layer had developed due to on-ice flow conditions. The spatial difference between the aircraft profiles and the dropsonde was about 10 km near the release altitude and about 50 km near the surface. To focus only on sensor-related differences and to minimise effects by spatial inhomogeneities of the surface, Tetzlaff (2016) selected measurements between 800–2000 m height for the comparison.

Tetzlaff (2016) found a very good agreement between the airborne and dropsonde measurements for wind despite the spatial differences. The average temperature measured with the dropsonde was 0.38 °C higher than during the aircraft profiles, which corresponded to twice the sensor's stated accuracy. The measured temperature variability was lower due to the lower sampling rate of the dropsonde's temperature sensor (Tetzlaff, 2016). Significant discrepancies occurred for relative humidity. The dropsonde data showed an absolute dry bias of 7 % to the reading of the relative humidity measured by aircraft (Tetzlaff, 2016). This corresponded with the value by Vance et al. (2004) derived based on a comparison of the dropsonde type used with radiosonde and airborne data. In addition, the dropsonde's GPS sensor overestimated the altitude of the sonde by about 25 m (Tetzlaff, 2016). For the subsequent analysis of the dropsonde measurements in the MCAOs during STABLE, Tetzlaff (2016) and also Tetzlaff et al. (2014) thus used the pressure- and temperature-based geopotential height instead of the less reliable GPS height. They also removed data between the dropsonde release and the points where the sensors adapted to the ambient conditions in the atmosphere. A correction of the dry bias for humidity was not considered by them.

### 4.2.1 Correction mechanisms with the ASPEN software

To provide a reliable, quality-controlled dropsonde data set, we applied most of the standard quality-processing mechanisms implemented in ASPEN (see Martin and Suhr, 2021) for correcting the AVAPS raw data. Most of the following procedures have also been applied in other studies, for example, by Vömel et al. (2021) for their dropsonde data set. The individual steps are as follows:

– Correction of the temperature-dependent dry bias for relative humidity using an algorithm provided by Vaisala (see also Vömel et al., 2016)

– Ambient equilibration for pressure, temperature, and relative humidity: All data between the release of the sondes and the time after the individual sensors had adapted to the environmental atmospheric conditions (equilibration) were skipped by this correction. The range of the corresponding data being discarded was calculated by specifying the equilibration time as seven times the individual sensors' time constants. Pressure and temperature equilibration times were set equal. For humidity, the largest equilibration time was applied since the humidity sensor's time constant is largest. By this correction, most of the data in the first 200–300 m from the launch point were discarded.

– Ambient equilibration for wind, where 10 s was set as equilibration time

– Post-splash check: The three dropsondes released over the MIZ (see Table 2 and Sect. 2.1) had still transmitted data after they had already reached the surface, which was detected by ASPEN automatically. The respective last relevant data points were then specified manually. All the other dropsondes released over open water did not contain post-splash data.

– Satellite check: A minimum of six satellites was set as a lower limit to ensure reliable GPS-based data (position, GPS height, fall velocity, and wind).

- Wind error check: Wind measurements for which the corresponding error was higher than a certain threshold value were discarded. The threshold values specified in ASPEN are $1.5\,\mathrm{m\,s^{-1}}$ for measurements between 100 m and 10 km altitude and $1\,\mathrm{m\,s^{-1}}$ for measurements below 100 m and above 10 km.

- Removal of data points outside predefined limits: Lower and upper limits are specified in ASPEN as 0 hPa and 1200 hPa for pressure, -100 °C and 50 °C for temperature, 0 % and 120 % for relative humidity, $0\,\mathrm{m\,s^{-1}}$ and $150\,\mathrm{m\,s^{-1}}$ for wind speed, 0 ° and 360 ° for the wind direction, and -300 m and 40000 m for the GPS altitude (adapted from https://ncar. github.io/aspendocs/algo_limitcheck.html).

- Removal of outliers: Outliers are treated as a specified multiple of the standard deviation $\sigma$ from a least-squares linear fit calculated for each data series. The multiples were specified as $4.5\sigma$ for pressure, $5\sigma$ for temperature, $10\sigma$ for relative humidity, and $5\sigma$ for the wind. These values correspond to the specifications suggested by the software developers for the dropsonde type used (see also https://ncar.github.io/aspendocs/algo_outlier.html).

- Removal of wild points: These points refer to single data points that differ by more than a certain threshold value (specified in terms of change per unit in time) with respect to the neighbouring points. Similar to the outliers, also these threshold values are user-specified, where we again chose the values suggested by the software developers. These are $1.5\,\mathrm{hPa \cdot s^{-1}}$ for pressure, $0.5\,\mathrm{°C \cdot s^{-1}}$ for temperature, $3\,\mathrm{\% \cdot s^{-1}}$ for relative humidity, $0.005\,\mathrm{deg \cdot s^{-1}}$ for GPS latitude and longitude, $50\,\mathrm{m\,s^{-1}}$ for GPS altitude, and $10\,\mathrm{m\,s^{-2}}$ for wind speed.

- Dynamic correction for wind and temperature: This procedure helped to diminish errors caused by the temperature sensor's time lag (see also Sect. 4.2.2) and by the sonde's inertia that affected the wind determination.

- Calculation of surface pressure and temperature by extrapolation using the lowermost data point and the sonde's fall rate

- Calculation of the geopotential height based on pressure, temperature, and relative humidity: ASPEN uses upward integration starting from the surface, or downward integration starting from the release altitude if the dropsonde had not transmitted data until reaching the surface. For our dropsonde data, all dropsondes transmitted data to the surface so that we applied upward integration only (see Sect. 4.2.3 for more details).

- Vertical (fall) velocity check: The reliability of the dropsonde's horizontal wind measurements can be well linked to its fall velocity measured by GPS. ASPEN compares the GPS-based fall velocity to the hydrostatically derived fall velocity and the theoretical fall velocity. They are calculated based on the time-differentiated hydrostatic equation or based on the sonde's and parachute's aerodynamic properties using a model, respectively. If the GPS-based fall velocity differed by more than $2.5\,\mathrm{m\,s^{-1}}$ from the other two velocities, the corresponding wind measurements were discarded.

- Calculation of the vertical wind component: This quantity was calculated based on the parachute's area and the parachute drag coefficient. We specified those as $0.09\,\mathrm{m^2}$ and 0.61, respectively, following Wang et al. (2009).

More detailed explanations of these quality-processing mechanisms are given by Martin and Suhr (2021).

The removal of data points that did not pass the quality-processing procedures by ASPEN predominantly concerned the uppermost dropsonde measurements and thus regions where the sensors had not yet adapted to the ambient conditions after the sondes' releases. In addition, a small number of data, namely just 2–5 s measurement time of all sondes, was removed by the vertical fall velocity check. A removal of wind measurements with wind error values exceeding the limits also mostly concerned the measurements just below the aircraft.

Some of the uppermost GPS-derived data of the sondes D5 from 4 March, D1 from 6 March, and D8 from 26 March were removed since the minimum number of satellites required for the calculations had not been reached. Hence, for the latter two sondes, GPS-derived data are not available for the uppermost 40–50 m after the pressure and temperature sensors had already adapted to the ambient conditions. For the sonde D5 from 4 March, GPS-derived data from the uppermost 200 m are missing in the quality-controlled data set since the reported number of GPS satellites was zero at that time. In almost all dropsonde data

series, a very small number of non-consecutive data points (mostly less than 10 per sonde) was removed by ASPEN since they were marked as having a cyclic redundancy check error, either for the pressure-temperature unit or for the GPS module.

### 4.2.2 Dry bias correction for humidity and dynamical adjustment

As expected, relative humidity values in the quality-controlled data are always higher than in the uncorrected data. Averaged over each data series, the correction ranges from +7.1 % to +9.9 % to the uncorrected relative humidity readings. These values

correspond with the above-mentioned dry bias values found in previous studies (Vance et al., 2004; Tetzlaff, 2016).

The dynamical adjustment applied to the temperature and wind measurements is the only procedure that actually modifies the measured data (Martin and Suhr, 2021). Regarding the temperature data, the adjustment leads to a seemingly faster response of the dropsondes' temperature sensors to strong small-scale changes with height. This helps to overcome the smaller temperature variability captured by the sondes as compared to the aircraft measurements (see also Tetzlaff, 2016). We illustrate this effect

in Fig. 3 for the three dropsondes that were released over sea ice. It is shown that the dynamical adjustment leads to more pronounced upper and lower boundaries of the temperature inversions, especially for those at a height of about 200–400 m, whereas the remaining measured temperature data are barely affected. For all dropsonde measurements, the average temperature of each profile changed by only about 0.05–0.25 K by the dynamical adjustment. The dynamical adjustment for wind caused hardly any change of the original wind values (not shown).

### 4.2.3 Geopotential height calculation

Since the three dropsondes that terminated over sea ice contained post-splash data, we can be sure that they transmitted data down to the surface. From the corresponding raw data series, we found, however, that the GPS-based height delivered values of about 25–35 m at the point where the surface was reached. This discrepancy is similar as described by Tetzlaff (2016). The other sondes transmitted data down to a GPS-derived height of about 25–50 m without containing post-splash data. This

means that all these sondes have most probably reached the surface although not indicated by the GPS-based height in the raw data. In addition, the sondes' measurements are transmitted at the end of the 0.5 s measurement cycle. This led to a loss

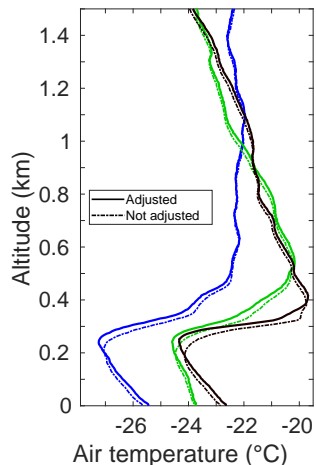

**Figure 3.** Air temperature data of the dropsondes D1 from 4 March (blue) and D1 (green) and D2 (dark red) from 26 March (see Table 2) with and without the dynamical adjustment of ASPEN (see Martin and Suhr, 2021). Only the lowest 1.5 km of the profiles are shown.

of data for the lowermost few metres for the sondes not reporting post-splash data (Hock and Franklin, 1999). By these two reasons, we can, however, assume that the corresponding sondes successfully transmitted data until reaching the sea surface. The sondes' surface data, which were then needed to perform the upward integration of the geopotential height, were computed
in ASPEN via extrapolation from the lowermost measurements transmitted. The upward integration was then performed until the uppermost quality-controlled pressure value was reached. Mostly, this is lower than the first few GPS-derived values of the wind by differences in the equilibration times assumed (Sect. 4.2.1). Thus, no geopotential altitude could be obtained for the uppermost few quality-controlled wind data points.

### 4.2.4 Wind speed error

As explained in Sect. 4.2.1, wind measurements exceeding a certain error threshold are discarded by ASPEN. In Fig. 4, we show the distribution of the wind error for the remaining quality-controlled wind data. It is shown that in most cases the median as well as the lower and upper quartiles of the wind error are between 0.4 and 0.7 m s$^{-1}$. Moreover, the whiskers in Fig. 4 denote that for all sondes approximately 99.7 % of the wind error values are between 0.4 and 0.9 m s$^{-1}$. However, for some data points, the wind error amounts to almost 1.5 m s$^{-1}$.

## 5 Horizontal distance

For model validation, the dropsondes' drift distances and the flight distances during the aircraft's descents/ascents might be important so that we provide this information in Fig. 5. Since we decided to plot the distances against the pressure-based altitude and not against the one based on GPS, the dropsondes' drift distances (Fig. 5a–d) are not shown for the layer where the meteorological sensors had not yet adapted to the environmental atmospheric conditions after release (see also Sect. 4.2.1).

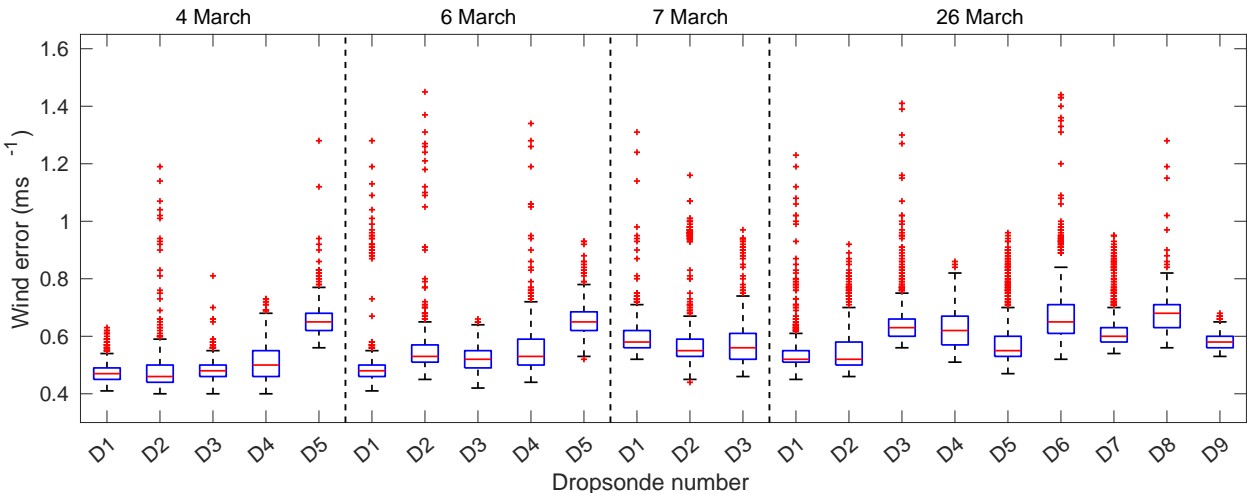

**Figure 4.** Distribution of the wind errors for each dropsonde listed in Table 2. Red lines show the median, the boxes are bounded by the upper and lower quartiles, the whiskers include the interval of approximately 99.7 % of the values, and red crosses denote outliers.

We also added the release altitudes of all sondes in Fig. 5a–d, where this information is only available from GPS. Most of the sondes showed a monotonic drift mainly along the main flow direction in the respective MCAO, except for D2 from 6 March. Especially on 26 March, some sondes drifted faster in the lowest kilometre of the atmosphere than at higher altitudes. This indicates a stronger wind speed in the convective ABL compared to the free atmosphere (see also Chechin et al., 2013). None of the sondes had been drifting more than 5 km, where the drift was much stronger on 4 March than on the other days.

Due to the small inclination of the aircraft during the vertical profiles (trajectory angles between -3° and -5° for descents and between 5° and 8° for ascents, on average), the travel distance and thus the horizontal distance of the measurements during one descent/ascent was much higher than for the dropsondes. During STABLE, these horizontal distances summed up to 30–40 km for an ascent from the surface to 3 km altitude (Fig. 5e–g). During the shorter profiles over the MIZ further north, the horizontal travel distance was mostly in the range 2–13 km.

## 6 Cold-air outbreaks

In this section, we focus on the meteorological characteristics of the MCAOs based on the quality-controlled airborne and dropsonde data. Figures 6 and 7 show vertical cross-sections of temperature, wind speed, as well as specific and relative humidity. Note that for the latter quantity, we show only those regions where saturation over ice was reached indicating where clouds had presumably been present. Panels of the total values of relative humidity are shown in Appendix B. All cross-sections are designed in a similar way as shown in Tetzlaff et al. (2014) for potential temperature and in Tetzlaff (2016) for all four variables. Data from the aircraft's profiles are the same in their and in our illustrations, whereas different corrections were applied by them to the dropsonde data (see Sect. 4.2.1). Figures 6 and 7 show also the ABL height on each day, which was also

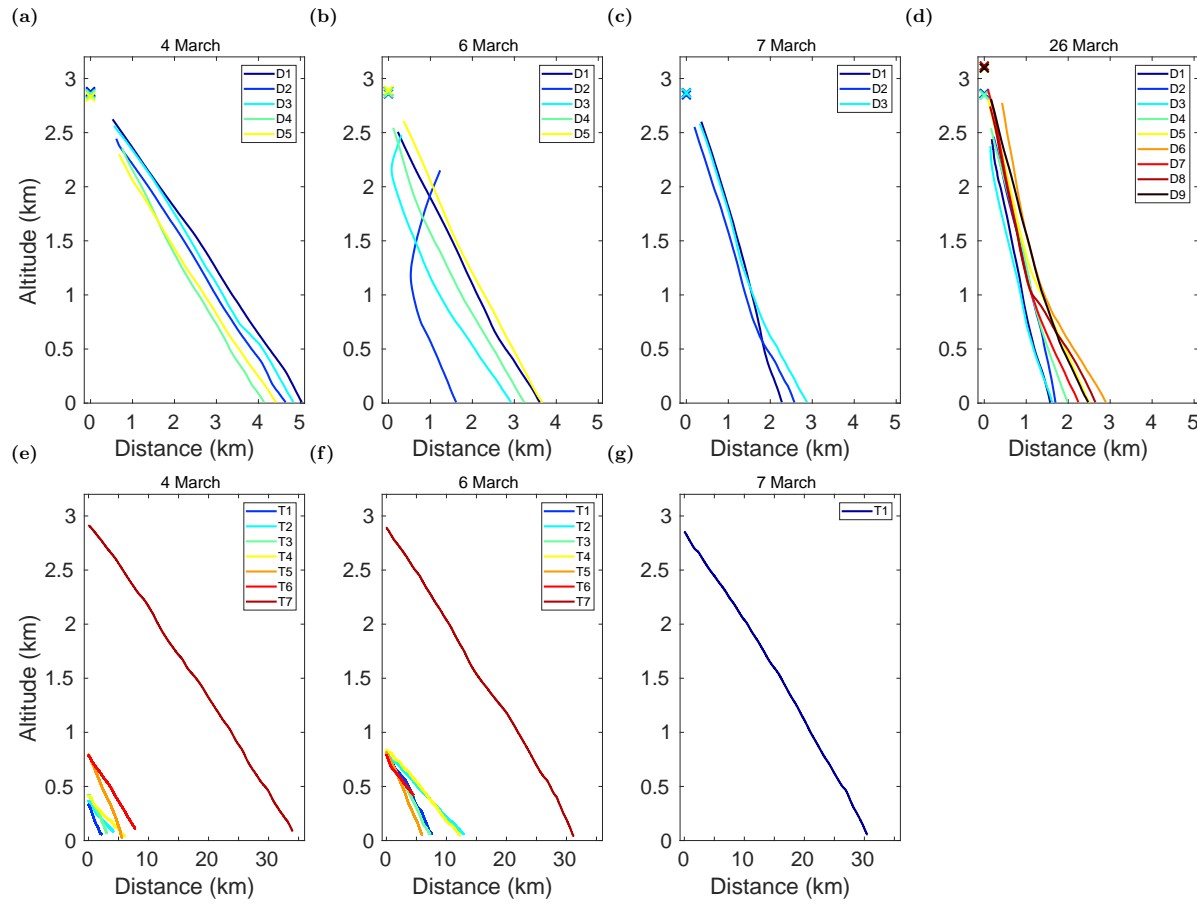

**Figure 5.** Horizontal drift distances of the dropsondes (a)–(d) and distances flown by the aircraft during ascents and descents (e)–(g). Distances are calculated with respect to the uppermost measurement point. For the dropsondes, this information is missing for the uppermost layer (see text), and coloured crosses refer to the dropsondes' release altitudes derived via GPS.

adapted from Tetzlaff (2016). We provide only a brief description of the cross-sections and refer to Tetzlaff et al. (2014) and Tetzlaff (2016) for a more detailed interpretation.

## 6.1 MCAO episode from 4 to 7 March

On 4 March, the observed MCAO was characterised by a shallow, slightly stably stratified ABL of about 250 m depth over sea ice and by a strong increase in both ABL height and temperature with increasing distance over open water (Fig. 6a). As noted by Tetzlaff et al. (2014), the resulting ABL height of about 2500 m at a distance of 214 km was extraordinarily high as compared to MCAOs observed in the early 1990's (e.g., Brümmer, 1997). A strong increase with increasing distance was also observed for both specific humidity and horizontal wind speed inside the ABL (Fig. 6c, e). The relative humidity distribution depicts clouds in the entire ABL over open water with increasing cloud base and top further to the south (Fig. 6g). Another

shallow cloud layer is denoted above the ABL at about 50 km distance. Moreover, it is shown that clouds were presumably also present over sea ice on that day, except for the northernmost flight legs. This agrees with visual observations made by the participants of the corresponding research flight.

On 6 March, the ABL over sea ice was slightly shallower than two days before. Moreover, the increase in both temperature and ABL height with increasing distance was not as pronounced as on 4 March (Fig. 6b). The specific humidity distribution denotes a slightly more humid ABL over sea ice but a similarly humid ABL over open water as compared to 4 March (Fig. 6d). Horizontal winds had basically decreased from 4 to 6 March, especially over open water (Fig. 6f). The relative humidity distribution denotes the presence of clouds over both sea ice and the open ocean. Compared to 4 March, less clouds might have

been present over sea ice on 6 March (Fig. 6h).

Unlike for the previous two days, no airborne measurements are available for the MIZ region on 7 March. The corresponding cross-sections indicate a weakening of the MCAO as compared to 4 and 6 March. Namely, specific humidity and horizontal wind speed in the ABL over open water were lower than on 4 and 6 March (Fig. 7a, c, e). Also, the increase of temperature and ABL height with increasing distance was less pronounced. The corresponding relative humidity distribution (Fig. 7g) denotes

several clouds not only in but also above the ABL along the entire cross-section.

## 6.2   MCAO event on 26 March

The measurements of the dropsondes D2–D5, which were released almost exactly along the main ABL flow direction in the MCAO, denote a deepening of the ABL from approximately 250 m near -50 km distance over the MIZ to 1250 m height at about 160 km distance over open water (Fig. 7b). This coincides with a potential temperature increase in the ABL from approximately

248 K to 258 K and a slight increase in both specific humidity and horizontal wind speed (Fig. 7b, d, f). Figure 7h denotes that clouds were presumably present in the entire ABL along the MCAO orientation and also above the ABL over the polynya region. In the region further downwind where the dropsondes D6–D9 were released, the ABL-averaged wind direction had turned slightly to north (see Fig. 1d), and the ABL had not deepened further with increasing fetch.

## 7   Data availability

The data are available on PANGAEA repository (Lüpkes et al., 2021a). All 37 data series can be found at https://doi.pangaea. de/10.1594/PANGAEA.936635 together with a brief data description. The individual DOIs of each data series are also listed in this paper in Tables 1 and 2. All data series are provided as tab-delimited text files (ascii). Missing or discarded measurements are indicated by empty fields in those files, where this only concerns the dropsonde data. All data are supplemented by metadata with information on the location and time of the measurements, the corresponding event, and on the measured quantities.

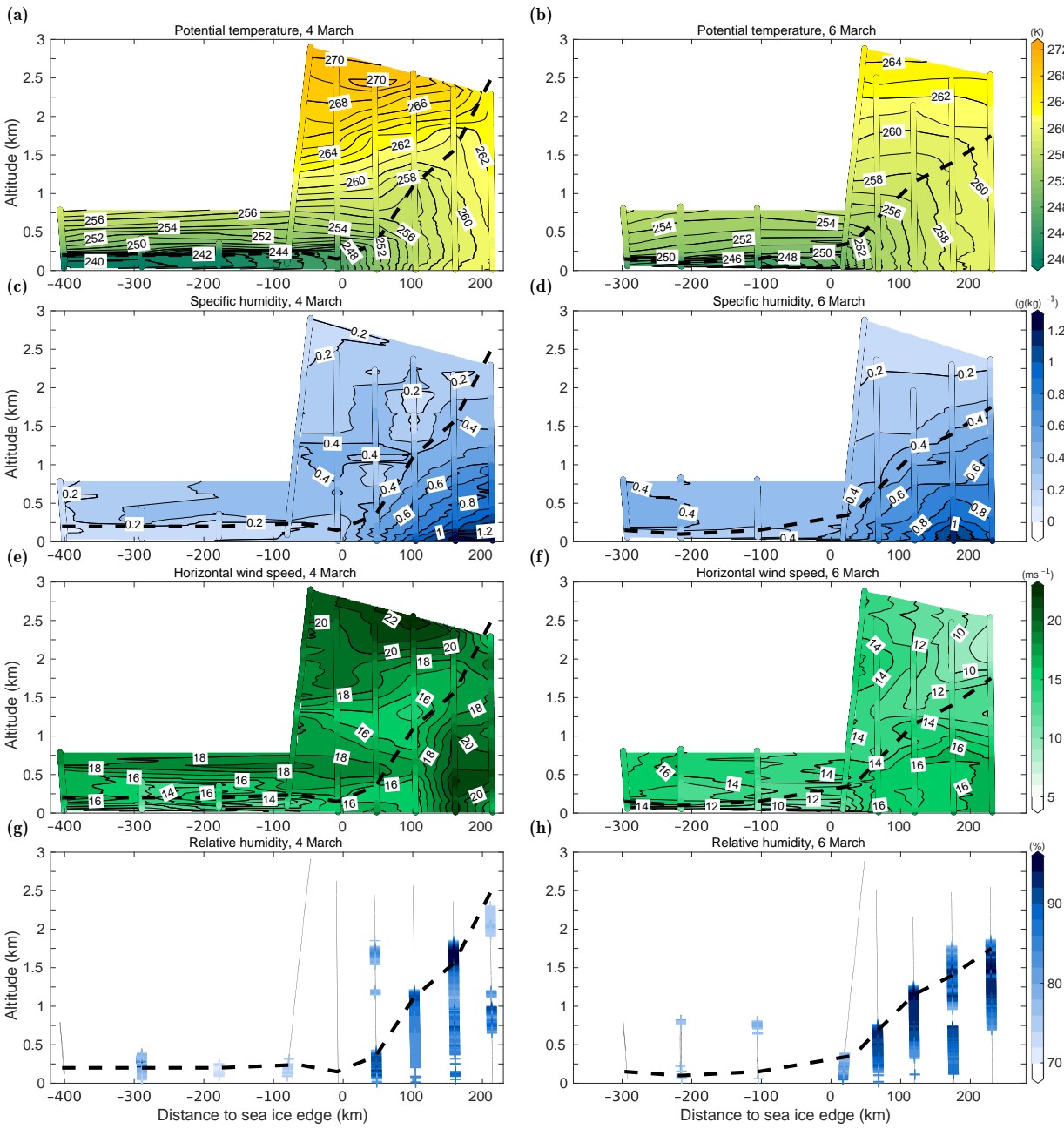

**Figure 6.** Vertical cross-sections of (a)–(b) potential temperature, (c)–(d) specific humidity, (e)–(f) horizontal wind speed, and (g)–(h) relative humidity in saturated areas with respect to an ice surface for (a), (c), (e), (g) 4 March and (b), (d), (f), (h) 6 March 2013 based on the quality-controlled airborne and dropsonde measurements. The flow is from left to right so that negative/positive distances correspond to sea ice/open water. Dashed black lines denote the ABL height. Modified based on Tetzlaff et al. (2014) and Tetzlaff (2016).

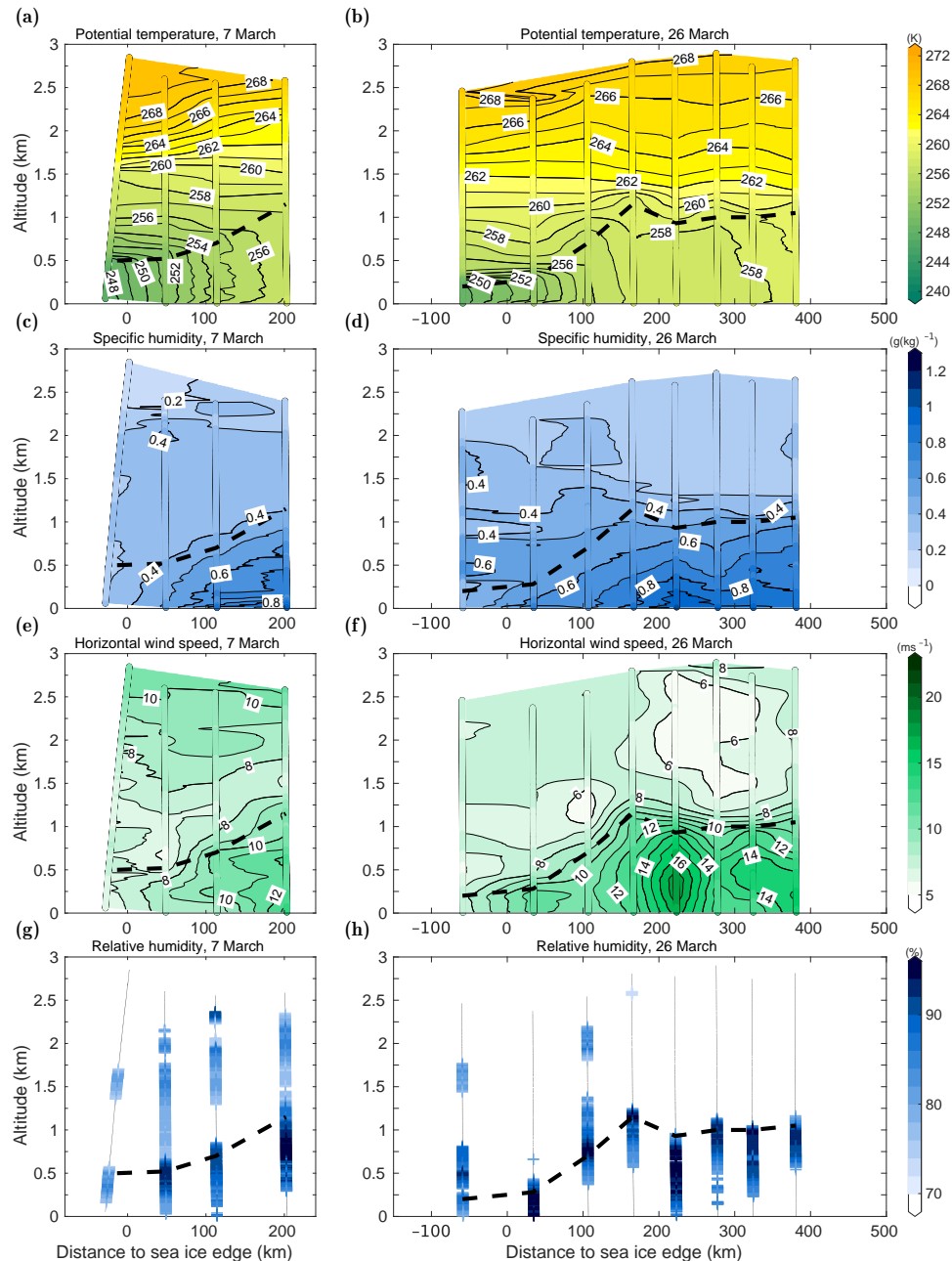

**Figure 7.** Same as Fig. 6, but (a), (c), (e), (g) show vertical cross-sections for 7 March and (b), (d), (f), (h) for 26 March 2013. Modified based on Tetzlaff et al. (2014) and Tetzlaff (2016).

# 8 Conclusions

The aircraft campaign STABLE was performed in March 2013. One of its main objectives was the investigation of atmospheric convection and boundary layer modifications associated with marine cold-air outbreaks (MCAOs). MCAOs occurring over the Fram Strait region were observed on four days using highly resolved atmospheric measurements from instruments mounted in and at the aircraft's nose-boom and by dropsondes. The observations took place in a period of an unusually large Whaler's Bay Polynya north of Svalbard, which had also led to extraordinarily deep convective boundary layers during the MCAOs only 200–250 km south of the ice edge (see Tetzlaff et al., 2014; Tetzlaff, 2016). We gave a detailed description of the corresponding data from, in total, 15 aircraft vertical profiles and 22 dropsondes. Data from the aircraft's profiles predominantly referred to observations over the marginal sea ice zone up to about 400 km upwind of the ice edge, whereas most of the dropsonde measurements took place over the open ocean. Thus, the research flights were arranged to allow detailed lower tropospheric observations ranging from the MCAOs' source regions far north of the ice margin to the open ocean region with the evolving convective boundary layers. Moreover, the flights followed approximately the mean wind direction.

To obtain a high-quality data set, we applied several quality-processing steps for both aircraft and dropsonde measurements. Especially for the dropsonde data, multiple corrections had to be applied. So, we used quality-processing algorithms implemented in the Atmospheric Sounding Processing Environment software (ASPEN, Martin and Suhr, 2021). This did not only help to remove suspicious data points but, for example, also to correct the dry bias found in the dropsonde measurements of relative humidity (see Vance et al., 2004). Moreover, a dynamical adjustment applied to the dropsondes' temperature measurements improved the representation of the observed temperature inversions and generally helped to overcome the reduction in the captured temperature variability as compared to the aircraft measurements.

Our data set refers to observations of one MCAO episode lasting for three days and to an MCAO that directly affected the Whaler's Bay Polynya. This brings up at least two aspects under which the data could be used in further studies. The first one belongs to the temporal and spatial variability of the MCAO episode from 4–7 March 2013. Second, the data could act as reference for investigations of MCAOs under an extreme scenario of springtime Arctic sea ice cover north of Svalbard. Thus, the data may be useful for micro- and mesoscale process-oriented modelling approaches (as, e.g., in Lüpkes and Schlünzen, 1996; Gryschka et al., 2008, 2014; Chechin et al., 2013) as well as for larger scale projections that assume such an extremely low ice concentration north of Svalbard as observed during STABLE.

Altogether, the data set we presented here consists of reliable and highly resolved atmospheric measurements of temperature, humidity, wind, and pressure. It provides a detailed representation of the vertical structure of the lower troposphere inside and above the evolving boundary layers during the MCAOs. Thus, it might serve as a valuable reference for comparisons with other observational data as well as for validation of model simulation results for such events of polar airmass transformations.

## Appendix A: Flight tracks

Figure A1 illustrates the flight tracks of the research flights from STABLE on 4, 6, 7, and 26 March 2013. On 4 and 6 March, the flight patterns mainly consisted of meridionally oriented flight legs in the region of the MCAOs (see also Sect. 2.1). On 7

March, additional measurements were performed during a low-level flight leg nearly parallel to the ice margin west of Svalbard. On 26 March, airborne measurements were performed over a lead located near 81.6° N and 21.4° E (see also Tetzlaff et al., 2015; Michaelis et al., 2021) prior to the measurements in the MCAO that occurred over the Whaler's Bay Polynya.

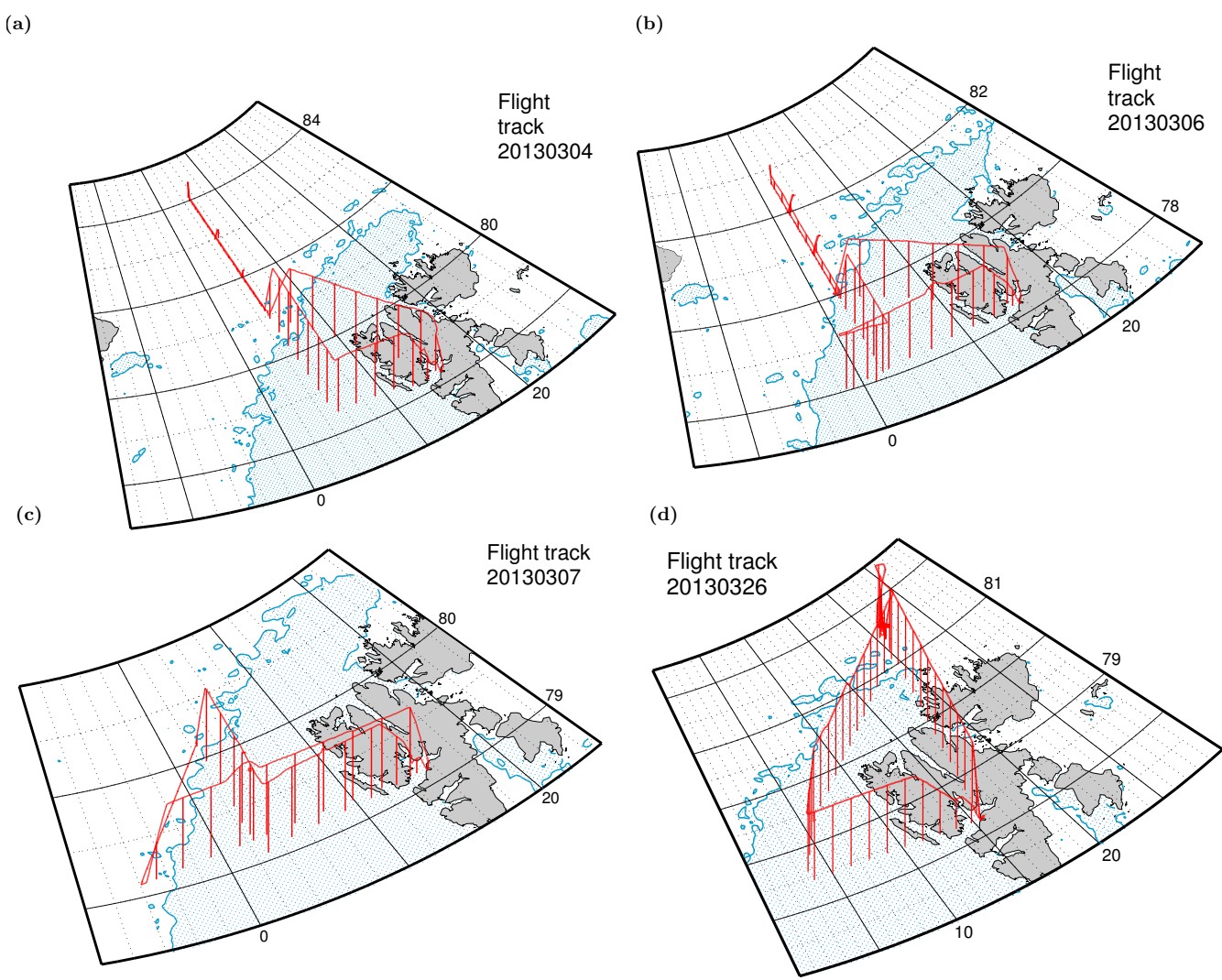

**Figure A1.** 3D illustration of the flight tracks on 4 March (a), 6 March (b), 7 March (c), and 26 March 2013 (d) plotted over the ice edge based on a 70 % threshold value of the ice concentration (see also Fig. 1 in Sect. 2). The length of the vertical lines between the flight tracks and the surface indicates the height of the aircraft during the research flights.

## Appendix B: Vertical cross-sections of relative humidity

Similar to Fig. 6g, h and Fig. 7g, h shown in Sect. 6, we provide in Fig. B1–B2 vertical cross-sections of the relative humidity based on the airborne and dropsonde measurements for 4, 6, 7, and 26 March 2013 in the respective MCAOs. Unlike the corresponding panels in Sect. 6, here we show the distribution of the total relative humidity. While Fig. 6g, h and Fig. 7g, h helped to detect regions of saturated air and thus clouds, Fig. B1 and Fig. B2 help to better identify dry and humid regions in the atmosphere nearly along the main MCAO orientations. For example, it is clearly shown that on 4 March, the region of the free atmosphere above the convective ABL over open water was basically much drier than on 6 March (Fig. B1a, b). Low relative humidity is also shown above the temperature inversion on 26 March, starting at a fetch of about 150 km (Fig. B1c).

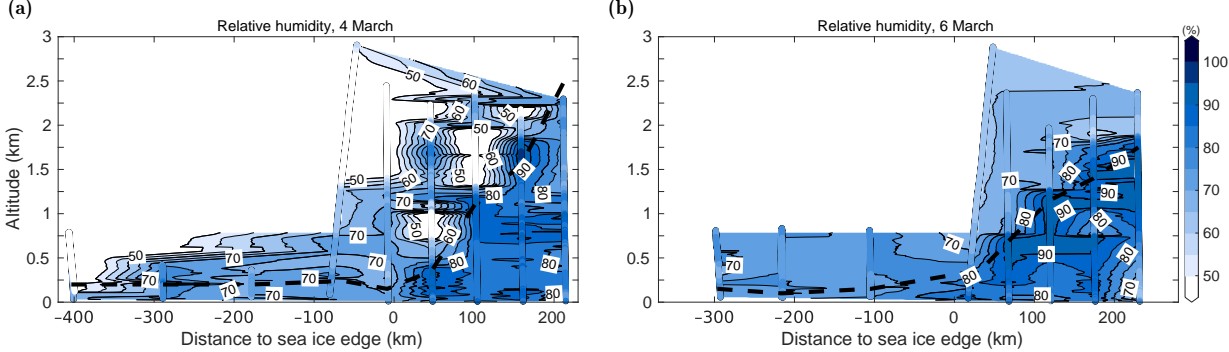

**Figure B1.** Same as Fig. 6, but vertical cross-sections are shown for the total values of relative humidity on (a) 4 March and (b) 6 March 2013. Modified based on Tetzlaff et al. (2014) and Tetzlaff (2016).

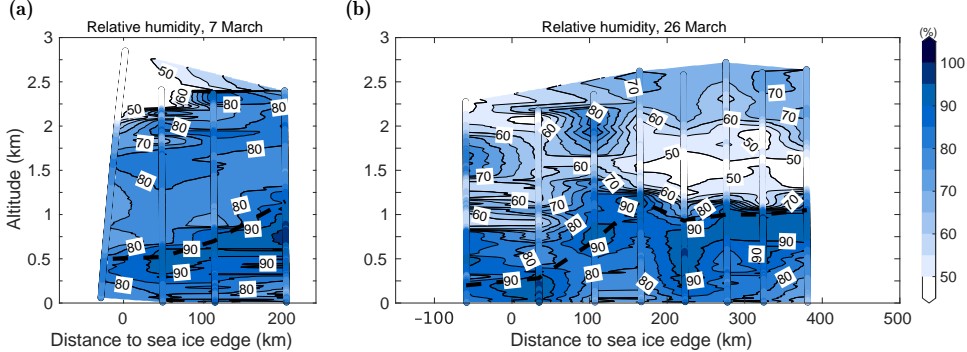

**Figure B2.** Same as Fig. 7, but vertical cross-sections are shown for the total values of relative humidity on (a) 7 March and (b) 26 March 2013. Modified based on Tetzlaff et al. (2014) and Tetzlaff (2016).

*Author contributions.* AS, CL, JH, GB, and TV all participated in planning the research flights during STABLE, and they operated the instruments onboard the aircraft including the AVAPS dropsonde system. The quality-processing was performed by JH for the aircraft data and by JM for the dropsonde data using the dropsonde software ASPEN. Data visualisation was done by JM with contributions from AS. JM performed the data analyses and prepared the manuscript. All co-authors contributed to the final writing of the paper.

*Competing interests.* The authors declare that they have no conflict of interest.

*Acknowledgements.* We gratefully acknowledge the funding by the Deutsche Forschungsgemeinschaft (DFG, German Research Foundation), project no. 268020496 – TRR 172, within the Transregional Collaborative Research Center: "ArctiC Amplification: Climate Relevant Atmospheric and SurfaCe Processes, and Feedback Mechanisms (AC)[3]". We are grateful to all aircraft engineers and the flight crew for their support during the campaign. We also thank Andreas Walbröl and Jan Chylik for helpful comments. Finally, we thank Ian Brooks and an anonymous reviewer for their constructive criticism that helped to improve the manuscript.

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
