# Peer review of "Observations of marine cold-air outbreaks: A comprehensive data set of airborne and dropsonde measurements from the Springtime Atmospheric Boundary Layer Experiment (STABLE)"

_Earth System Science Data, 2021_

## Referee Comment (RC2)

Review of ESSD-2021-341: "**Observations of marine cold-air outbreaks: A comprehensive data set of airborne and dropsonde measurements from the Springtime Atmospheric Boundary Layer Experiment (STABLE)**" by Janosch Michaelis, Amelie U. Schmitt, Christof Lüpkes, Jörg Hartmann, Gerit Birnbaum, and Timo Vihma.

Ian Brooks

**Overview**

This paper provides an overview and brief description of (near) vertical profile measurements in cold air outbreaks in the vicinity of the Fram Strait. The data consist of profiles of mean meteorological data (temperature, winds, humidity), along with position and altitude, from dropsondes and aircraft profiles. They are a subset of measurements from an airborne measurement campaign focused on boundary layer structure.

The paper is generally clear and well written, requiring only minor revision before publication.

**Detailed Comments**

The introduction focuses primarily on a brief background of cold air outbreaks, and some of the specifics of the papers by Tetzlaff et al. – which used the data documented here – before finally introducing the data and purpose of this paper. This is fairly typical for a science paper, but seems a little awkward for a data paper, where the data set might be better introduced first, before recapping published science using it, and the science areas it is aimed at supporting.

Line 44: "quality-processed" might be better changed to "quality-controlled" (here and elsewhere)

Line 116: "one dropsonde at the same time" -> "only one dropsonde at a time"

Line 123: "a spatial resolution" -> "a vertical resolution" – be explicit that it is vertical resolution here, 'spatial' could be read as implying horizontal resolution.

Line 153: "twice the sensor's accuracy" -> "twice the sensor's stated accuracy" – there is a distinction to be made here between the stated accuracy from the manufacturer, and the actual accuracy of the measurement, which is found to be rather less than that stated.

Line 155: "the used dropsonde type" -> "the dropsonde type used"

Line 165-196: it would be useful to include within this list the values for the various thresholds etc used to exclude data. Some are given, and it is implied that the others can be found in the cited papers, but it would be a useful reference to have them all listed together in one place here.

Section 4.1: The discussion of quality control & corrections applied to the aircraft data is somewhat limited. The discussion of GPS altitude errors is fine, and does a good job of explaining these. The discussion of temperature corrections is very brief – though perhaps there isn't a lot more to say. There is no mention here of humidity measurements.
It might make things a little easier to follow – and would certainly help for simple reference – to merge the description of QC-processing from section 3, with the effects it has in section 4: structure by aircraft/dropsonde rather than QC-processing-methods/impacts.

Line 204: 'subsequently' seems an odd, and unnecessary, start to the sentence.

Line 205: 'temperatures are generally lower after the correction'…only 'generally' (ie some are higher), or are they not always lower since the dynamic pressure should always be positive?

Line 258-259: 'are not shown for the layer where the meteorological sensors had adapted to the environmental atmospheric conditions…' – shouldn't this be '…sensors had not adapated…' ?

Line 265-266: 'the aircraft had a pitch angle of not more than about ±2–±10°' – two issues here. First the statement of the angle is a little confusing as the dash is easily read as a minus sign or the ±10 as an uncertainty about a range of ±2. Is the intended meaning '±2° to ±10°' – ie between 2 and 10 degrees up or down and thus excluding angles between ±2?
Second point – this range is stated as being the aircraft 'pitch' (angle of orientation wrt horizontal) and it is then stated that the small inclination means the aircraft travels a significant horizontal distance during its ascent/descent profiles. The relevant angle for the horizontal distance travelled is not the aircraft's pitch, but that of its trajectory…it can be descending while pitched upwards.

---

## Author Comment (AC1)

**Author responses to reviewer comments**

We thank both reviewers for taking their time to review our manuscript and for the valuable comments. Below, we mention the reviewers' comments in italic letters and add then our comments.

**Response to RC1 ('Comment on essd-2021-341', Anonymous Referee #1)**

1. *The authors describe in this paper the quality control of dropsonde and aircraft measurements, done in the STABLE experiment. Additionally, a brief overview about the scientific results of the measurements are given. Even if I miss only some details, I ask the authors to revise the paper in order to group by instruments. Now, a description of the instruments (subsections for aircraft and dropsondes) is followed by a section of the quality-processing, again with subsection for the aircraft and dropsondes. Next section is about data quality and statistics with subsections for the aircraft and dropsondes, where the dropsonde subsection has several subsubsections. I propose to have after the introduction a chapter with sections about the aircraft measurements, their post-processing and data quality and same for the dropsondes. I hope this avoids the frequent turn the pages while reading the article.*
   *Another general point are the very long sentences. Sometimes, several thoughts are within one long sentence. I propose to split the long sentences and have one thought per sentence.*

   Thank you for pointing to all this. The comment has two aspects. Regarding the first one, we rearranged the order of the sections after the introduction and before the cold air outbreaks-section (which is now section 6) following the reviewer's suggestions. First, we rearranged section 2 of the original manuscript so that it now contains the general description of the campaign STABLE and the information on the flight patterns (the former section 2.1). Then, in the new section 3, we describe now the aircraft measurements, followed by the dropsonde measurements in section 4. Both sections now contain a description of the respective instruments (sections 3.1 and 4.1) and of the quality-processing and its effects (sections 3.2 and 4.2). Section 4.3 of the original manuscript, which contained information of the horizontal distances covered during the individual airborne and dropsonde measurements, was moved to section 5 in the revised manuscript (see also our answer to reviewer comment #11). The cold air outbreaks are now described in section 6, followed by the data availability statement in section 7, and the conclusions in section 8.

   Regarding the second point, we thoroughly considered the whole text. Very long sentences were either just shortened or partitioned into two sentences, whenever it was possible.

2. *Figure 1, 3rd line: "Squares in (a)-(c)" -> I cannot see any square in the maps.*

   The squares showing the positions of the aircraft vertical profiles become visible when zooming into the figure. However, we noticed that they are difficult to distinguish from the circles showing the dropsondes' positions. Hence, we modified the caption of Figure 1.

3. *Line 125: The accuracy is 2 % for the relative humidity. Is it 2 % of the current reading or +/-2 % for all readings (2 % accuracy and 50 % relative air humidity can be 50+/-2 % or 50+/-1 %).*

   The accuracy for relative humidity is quoted by the manufacturer as "2 % RH" so that it is $\pm 2\,\%$ for all readings. We added this information in the revised manuscript (see section 4.1).

4. *Line 135: How are outliers defined? How are these corrected?*

   We clarify this now in section 3.2 of the revised manuscript. There, we write now: "Although there was not a standard procedure to remove spikes or outliers after the basic processing as described in Hartmann et al. (2018) had been applied, all data series shown here were inspected visually. Sections of aircraft data where invalid values have been identified, for example, due to the influence by icing, are not included in the data stored in the repository."

5. *Line 137: How was the correction of air pressure data done?*

   We have added a more detailed description of this to section 3.2. There, we write now "Air pressure data were corrected for the influence of the flow field around the aircraft. The corrected (static) air pressure $p_s$ was obtained via

$$p_s = p_{s,i} + q_i(1 - c) + \Delta p_s, \tag{1}$$

   where $p_{s,i}$ is the uncorrected static pressure, $q_i$ is the uncorrected dynamic pressure, $c$ is a calibration factor to obtain the corrected dynamic pressure (with $c = 1.165$), and $\Delta p_s$ is the measurement error of the five hole probe depending on the flow angle (see Hartmann et al., 2018, their Eq. (6)). The constant $c$ had been obtained by Hartmann et al. (2018) from several pairs of reverse-heading flight sections during which the mean wind had changed only little.

6. *Line 155: Is it an absolute or relative bias?*

   The dry bias of $7\,\%$ is an absolute bias. We clarified this in the revised manuscript. Moreover, for comparison with the dry bias found in the cited literature, we briefly mentioned the dry bias we found in our dropsonde data set (range is 7.1–9.9 % RH, see section 4.2.1 in the revised manuscript).

7. *Line 179: "...corresponding error was higher than..." -> How were these errors calculated?*

   We have added this to the first paragraph of section 4.1, wherein we describe the instrumentation of the dropsondes. There, we write now: "Following Hock and Franklin (1999), the wind error is determined based on the measurement errors in the sondes' horizontal and vertical velocities and accelerations. The latter error consists of a random component (i.e. noise in the velocity estimates) and of a sampling component due to the sampling interval of the wind measurements (see Hock and Franklin, 1999)."

8. *Line 183: How were the predefined limits defined? How were outliers defined? How were wild points defined?*

   We have added the corresponding values as well as more detailed explanations of the definitions in section 4.2.1 of the revised manuscript.

9. *Lines 193/194: "...differed too much..." -> How much is too much?*

   The corresponding vertical velocity difference limit is specified as $2.5\,\mathrm{m s}^{-1}$. We added this information in the revised manuscript (see the second last bullet point of the list in section 4.2.1).

10. *Section 4.1: I miss the quality-processing of the wind data.*

    We clarified this now in the new section 3.2 of the revised manuscript. There, we write now: "The wind components were calculated by the difference between the aircraft's velocity and the vector of the true airflow following the method described in detail by Hartmann et al. (2018). While the former component was obtained with a high accuracy from the GPS and INS, the latter was obtained from the quality-controlled pressure measurements. Thus, for the wind components, there was no additional quality-processing necessary apart from the manual check of all time series from which invalid sections had been removed." Please consider also our answers to reviewer comments #4 and #5.

11. *Lines 265-268: Why are aircraft measurements in the dropsonde section? Please move it to the aircraft section.*

    As already mentioned in the answer to the reviewer's first comment, we have rearranged the sections of the manuscript's main body so that the aircraft measurements are now described first (section 3) followed by the dropsonde measurements (section 4). However, we still want to keep the information of the horizontal distance (by the dropsondes' drift and by the aircraft's flight distances during the individual legs) in a single figure. Therefore, we moved the content of the former subsection 4.3 ("Horizontal distance") into an own section after the description of aircraft and dropsonde measurements (see section 5 in the revised manuscript).

12. *Figures 6 & 7: Indicate, which distances correspond to sea ice and which to open water.*

    Positive distances correspond to open water and the negative ones to sea ice. We added this information in the caption of Figure 6.

**Response to RC2 ('Comment on essd-2021-341', Ian Brooks)**

**Overview**

*This paper provides an overview and brief description of (near) vertical profile measurements in cold air outbreaks in the vicinity of the Fram Strait. The data consist of profiles of mean meteorological data (temperature, winds, humidity), along with position and altitude, from dropsondes and aircraft profiles. They are a subset of measurements from an airborne measurement campaign focused on boundary layer structure. The paper is generally clear and well written, requiring only minor revision before publication.*

**Detailed Comments**

1. *The introduction focuses primarily on a brief background of cold air outbreaks, and some of the specifics of the papers by Tetzlaff et al. – which used the data documented here – before finally introducing the data and purpose of this paper. This is fairly typical for a science paper, but seems a little awkward for a data paper, where the data set might be better introduced first, before recapping published science using it, and the science areas it is aimed at supporting.*

   Thank you for pointing to this. We have restructured the introduction according to the suggestions.

2. *Line 44: "quality-processed" might be better changed to "quality-controlled" (here and elsewhere)*

   We replaced "quality-processed" by "quality-controlled" at every occurrence in the manuscript.

3. *Line 116: "one dropsonde at the same time" -> "only one dropsonde at a time"*

   We changed this.

4. *Line 123: "a spatial resolution" -> "a vertical resolution" – be explicit that it is vertical resolution here, 'spatial' could be read as implying horizontal resolution.*

   Thanks for pointing to this. We replaced "spatial resolution" by "vertical resolution".

5. *Line 153: "twice the sensor's accuracy" -> "twice the sensor's stated accuracy" – there is a distinction to be made here between the stated accuracy from the manufacturer, and the actual accuracy of the measurement, which is found to be rather less than that stated.*

   We thank for this important remark and added the word "stated" to the corresponding sentence.

6. *Line 155: "the used dropsonde type" -> "the dropsonde type used"*

   We changed this.

7. *Line 165-196: it would be useful to include within this list the values for the various thresholds etc used to exclude data. Some are given, and it is implied that the others can be found in the cited papers, but it would be a useful reference to have them all listed together in one place here.*

   We added the upper and lower limits as well as threshold values concerning the removal of values outside predefined limits, outliers, and wild points in the revised manuscript. We also

specified the threshold value used for the vertical fall velocity check. All this is now described in section 4.2.1 of the revised manuscript.

8. *Section 4.1: The discussion of quality control & corrections applied to the aircraft data is somewhat limited. The discussion of GPS altitude errors is fine, and does a good job of explaining these. The discussion of temperature corrections is very brief – though perhaps there isn't a lot more to say. There is no mention here of humidity measurements. It might make things a little easier to follow – and would certainly help for simple reference – to merge the description of QC-processing from section 3, with the effects it has in section 4: structure by aircraft/dropsonde rather than QC-processing-methods/impacts.*

First, we agree with the reviewer that it would be easier to follow the content when the description of QC-processing and its effects are merged for each of the instruments (aircraft/dropsonde). Therefore, following also the suggestion from reviewer #1, we have restructured the main part of the manuscript as follows: Section 2 now contains a brief overview of the campaign STABLE and of the flight patterns. Then, in the new section 3, we describe now the aircraft measurements, followed by the dropsonde measurements in section 4. Both sections now contain a description of the respective instruments (sections 3.1 and 4.1) and of the quality-processing and its effects (sections 3.2 and 4.2). Section 4.3 of the original manuscript, which contained information of the horizontal distances covered during the individual airborne and dropsonde measurements, was moved to section 5 in the revised manuscript.

Second, regarding the QC-processing of the aircraft data, we have added a more detailed description of the standard quality-controlling procedures and of the correction of the air pressure measurements in section 3.2. This has been made following also the suggestions by reviewer #1.

Third, regarding the humidity measurements, we did two things. First, we added in section 3.2 that there was no quality-processing necessary for the relative humidity measurements apart from the manual check of all time series from which invalid sections had been removed. Second, for the dropsondes' humidity measurements, we added information on the dry bias that was shown in the raw dropsonde data (see section 4.1 in the revised manuscript). There, we write now: "As expected, relative humidity values in the quality-controlled data are always higher than in the uncorrected data. Averaged over each data series, the correction ranges from $+7.1\,\%$ to $+9.9\,\%$ to the uncorrected relative humidity readings. These values correspond with the above-mentioned dry bias values found in previous studies (Vance et al., 2004; Tetzlaff, 2016)."

9. *Line 204: 'subsequently' seems an odd, and unnecessary, start to the sentence.*

We deleted the word "Subsequently".

10. *Line 205: 'temperatures are generally lower after the correction'…only 'generally' (ie some are higher), or are they not always lower since the dynamic pressure should always be positive?*

Thank you for this remark. Of course, temperatures are always lower after applying the correction for adiabatic heating. Thus, we skip the word "generally" in the revised manuscript.

11. *Line 258-259: 'are not shown for the layer where the meteorological sensors had adapted to the environmental atmospheric conditions...' – shouldn't this be '...sensors had not adapted...'?*

    This is correct. We added "not" to the corresponding sentence.

12. *Line 265-266: 'the aircraft had a pitch angle of not more than about $\pm2$–$\pm10°$' – two issues here. First the statement of the angle is a little confusing as the dash is easily read as a minus sign or the $\pm10$ as an uncertainty about a range of $\pm2$. Is the intended meaning '$\pm2°$ to $\pm10°$' – ie between 2 and 10 degrees up or down and thus excluding angles between $\pm2$? Second point – this range is stated as being the aircraft 'pitch' (angle of orientation wrt horizontal) and it is then stated that the small inclination means the aircraft travels a significant horizontal distance during its ascent/descent profiles. The relevant angle for the horizontal distance travelled is not the aircraft's pitch, but that of its trajectory...it can be descending while pitched upwards.*

    Thank you a lot for pointing to this. Regarding the first point, we meant that the pitch angle was between 2 and 10 degrees up or down on average for the individual vertical flight legs. Regarding the second point, we agree that the angle of the aircraft's trajectory is the relevant angle for the horizontal distance travelled. Thus, we changed the text in section 5 as follows: "Due to the small inclination of the aircraft during the vertical profiles (trajectory angles between -3° and -5° for descents and between 5° and 8° for ascents, on average), the travel distance and thus the horizontal distance of the measurements during one descent/ascent was much higher than for the dropsondes."

---

## Author Response (AR2)

**Response to referee report from Ian Brooks**

We thank the reviewer for taking the time to review our manuscript once again and for the valuable comments. Below, we mention the reviewer's comments in italic letters and add then our comments.

*All the reviewer comments from the initial submission appear to have been addressed satisfactorily. A few minor technical corrections are required here:*

1. *Throughout the manuscript, there should be a space between 'm' and 's' in the units for velocity.*

   We corrected this.

2. *line 140: 'only little' -> 'only a little'*

   We changed this.

3. *line 152: 'is at 2-3 K' -> 'is 2-3 K'*

   We corrected this.

4. *line 212: 'most of the data between the first 200–300m' -> 'most of the data in the first 200–300m'*

   We corrected this.

5. *line 288: 'uppermost few quality-controlled wind data.' -> 'uppermost few quality-controlled wind data points.'*

   We corrected this.

6. *Figure A1. I presume the vertical lines from the aircraft tracks to the surface are the dropsonde paths, but this isn't mentioned in either the figure caption or main text. Add text to make this clear."*

   The vertical lines do not correspond to the dropsonde paths. They have only been added to indicate the height of the flight tracks. The longer the vertical line is, the higher the aircraft has flown at that position. We added this information in the caption of Figure A1.